# Monoclonal Antibodies Can Aid in the Culture-Based Detection and Differentiation of *Mucorales* Fungi—The Flesh-Eating Pathogens *Apophysomyces* and *Saksenaea* as an Exemplar

**DOI:** 10.3390/antib14040085

**Published:** 2025-10-07

**Authors:** Christopher R. Thornton, Genna E. Davies

**Affiliations:** 1ISCA Diagnostics Ltd., Lowin House, Tregolls Road, Truro, Cornwall TR1 2NA, UK; 2Living Systems Institute, Stocker Road, Exeter, Devon EX4 4QD, UK; g.davies@exeter.ac.uk

**Keywords:** *Mucorales*, *Apophysomyces*, *Saksenaea*, mucormycosis, monoclonal antibody, antigen, lateral-flow immunoassay, ELISA

## Abstract

Background: The frequency of necrotising cutaneous and soft tissue infections caused by the *Mucorales* fungi *Apophysomyces* and *Sakasenaea* is increasing. The absence of sophisticated diagnostic technologies in low- and middle-income countries (LMICs) means that detection of cutaneous mucormycosis continues to rely on culture of the infecting pathogens from biopsy and their differentiation based on morphological characteristics. However, *Apophysomyces* and *Sakasenaea* are notorious for their failure to sporulate on standard mycological media used for the identification of human pathogenic fungi. Differentiation of these pathogens and their discrimination from *Aspergillus fumigatus*, the most common mould pathogen of humans, is essential due to their differing sensitivities to the antifungal drugs used to treat mucormycosis. Methods: A murine IgG1 monoclonal antibody, JD4, has been developed that is specific to *Apophysomyces* species. In Western blotting and enzyme-linked immunosorbent assay (ELISA), mAb JD4 is shown to bind to an extracellular 15 kDa protein, readily detectable in crude antigen extracts from non-sporulating cultures of *Apophysomyces*. Results: When combined with a *Mucorales*-specific lateral-flow immunoassay (LFIA), mAb JD4 allows the differentiation of *Apophysomyces* from *Saksenaea* species and discrimination from *Aspergillus fumigatus*. Monoclonal antibody JD4 enables the detection and differentiation of *Apophysomyces* species from other fungal pathogens that cause rapidly progressive cutaneous and soft tissue mycoses in humans. When this is combined with a rapid LFIA, improvements are offered in the sensitivity and specificity of *Mucorales* detection based on mycological culture, which remains a gold-standard procedure for mucormycosis detection in LMICs lacking access to more sophisticated diagnostic procedures.

## 1. Introduction

Mucormycosis is a lethal angio-invasive disease of humans caused by fungi in the order *Mucorales*, recently identified as a high-priority pathogen group [1,2,3]. The *Mucorales* fungi most commonly reported as human pathogens are species of *Lichtheimia*, *Mucor*, and *Rhizopus*, followed by *Apophysomyces*, *Cunninghamella*, *Rhizomucor*, and *Saksenaea* [4,5]. *Apophysomyces* spp. are the second most common agents of rhino-orbital-cerebral mucormycosis (ROCM) in India after *Rhizopus arrhizus* [6,7,8] and are responsible, along with *Saksenaea* spp., for the majority of cases of cutaneous and soft tissue mucormycosis in the form of necrotising fasciitis [9,10,11,12,13,14]. As emerging pathogens of humans [15,16], reports of *Apophysomyces* spp. in necrotising cutaneous and soft tissue infections of immunocompetent individuals following penetrating trauma and burns [4,13,17,18,19,20,21,22,23,24,25] have increased over recent years, particularly following natural disasters such as tornadoes and tsunamis, and following combat-related blast injuries [26,27,28,29,30].

The prompt detection and differentiation of *Mucorales* spp. from other mould pathogens such as *Aspergillus fumigatus* is critical to patient survival, enabling timely treatment with surgery and *Mucorales*-active antifungal drugs [4]. Detection of *Mucorales* pathogens has traditionally relied on fungal culture from a biopsy sample and identification by microscopy of characteristic morphological structures. However, *Apophysomyces* and *Saksenaea* species are notorious for their failure to sporulate on routine isolation media used for the cultivation of filamentous fungi [9,11,31,32], instead requiring lengthy and laborious methods to induce sporulation for identification by specialist medical mycologists [33,34,35,36,37]. For this reason, more sophisticated and costly diagnostic modalities such as DNA sequencing or matrix-assisted laser desorption/ionisation time of flight (MALDI-TOF) are used for identification following culture in vitro [11,38,39], but widespread access to such technology is limited, especially in low- and middle-income countries (LMICs). Such countries continue to rely on culture for the detection of fungal infections, which remains a cornerstone for definitive diagnosis of mucormycosis worldwide [4,14]. Indeed, the majority of cases of mucormycosis during the second wave of the COVID-19 pandemic in India were diagnosed by culture of the infecting fungi [40,41].

Antigens that can act as biomarkers for *Mucorales* fungi have recently been reported, including extracellular polysaccharides (EPSs) detectable using species-specific [42] and pan-*Mucorales*-specific [43] monoclonal antibodies (mAbs) in lateral-flow immunoassay (LFIA) and in immunohistochemistry [42,43,44,45].

In this paper, we report the development of a murine IgG1 mAb, JD4, which binds to a major 15 kDa antigen specific to the genus *Apophysomyces*. We show how the mAb can be employed in an enzyme-linked immunosorbent assay (ELISA) to differentiate *Apophysomyces* spp. from *Saksenaea* spp., following rapid discrimination of *Mucorales* pathogens from *Aspergillus fumigatus* using a simple culture-based antigen extraction procedure and *Mucorales*-specific LFIA.

## 2. Materials and Methods

### 2.1. Ethics Statement

The hybridoma work described in this study was conducted under a UK Home Office Project License and was reviewed by the institution’s Animal Welfare Ethical Review Board (AWERB) for approval on 20 January 2022. The work was carried out in accordance with The Animals (Scientific Procedures) Act 1986 Directive 2010/63/EU and followed all the Codes of Practice which reinforce this law, including all elements of housing, care, and euthanasia of the animals.

### 2.2. Fungal Culture

Fungi (Table 1) were routinely cultured on malt extract agar (MEA; 70145, Sigma, Poole, UK). The medium was autoclaved at 121 °C for 15 min prior to use, and fungi were grown at 37 °C. To induce sporulation in *Apophysomyces* spp., the fungi were grown on autoclaved Czapek Dox agar (70185, Sigma, Poole, UK) at 37 °C. To induce sporulation of *Saksenaea* spp., the method of Padhye and Ajello [34] was used. Briefly, blocks of mycelium were cut aseptically from seven-day-old colonies of *Saksenaea* spp. grown on Sabouraud dextrose agar (Sigma, 1468430020, Poole, UK) and transferred to 9 cm diameter culture plates containing 20 mL of sterile distilled water (dH_2_O) supplemented with 0.2 mL of a 10% (vol:vol) solution of filter-sterilised yeast extract (Sigma, Y1625, Poole, UK). The plates were then incubated for 10 d at 37 °C. This resulted in the abundant production of spore-bearing sporangia.

### 2.3. Production of Hybridomas and Screening by ELISA

#### 2.3.1. Hybridoma Production

Extracellular polysaccharides (EPSs) were prepared according to Davies and Thornton [42]. For hybridoma production, the immunogen comprised a 1 mg/mL solution of EPS from *Apophysomyces variabilis* CBS658.93, with four six-week-old female BALB/c white mice each given four intra-peritoneal injections (300 µL per injection) of immunogen at 2 wk intervals and a single booster injection 5 d before fusion. Hybridoma cells were produced by a method described elsewhere, and monoclonal antibody (mAb)-producing clones were identified in Indirect-ELISA tests [46].

#### 2.3.2. Indirect-Enzyme-Linked Immunosorbent Assay

For Indirect-ELISA, EPS at a concentration of 20 µg/mL phosphate-buffered saline (PBS; 137 mM NaCl, 2.7 mM KCl, 8 mM Na_2_HPO_4_, 1.5 mM KH_2_PO_4_ [pH 7.2]) was immobilised to the wells of two replicate Maxisorp microtiter plates (Nunc) at 50 µL/well. The wells containing immobilised antigen were incubated with 50 µL of mAb hybridoma tissue culture supernatant (TCS) for 1 h, after which the wells were washed three times, for 5 min each, with PBST (PBS containing 0.05% (vol:vol) Tween-20). Goat anti-mouse polyvalent immunoglobulin (G, A, M) peroxidase conjugate (PA1-84388; Invitrogen, Loughborough, UK), diluted 1:5000 (vol:vol) in PBST, was added to the wells and incubated for a further hour. The plates were washed with PBST as described, then given a final 5 min wash with PBS. Bound antibody was visualised by incubating wells with tetramethyl benzidine (TMB) substrate solution [46] for 30 min, after which reactions were stopped by the addition of 3 M H_2_SO_4_. Absorbance values were determined at 450 nm using a microplate reader (infinite F50, Tecan Austria GmbH, Reading, UK). Control wells were incubated with tissue culture medium (TCM) containing 10% (vol:vol) foetal bovine serum (F7254, Sigma, Poole, UK) only. All incubation steps were performed at 23 °C in sealed plastic bags. The threshold for detection of the antigen in Indirect-ELISA was determined from control means (2 × TCM absorbance values). These values were consistently in the range of 0.050–0.100. Consequently, absorbance values ≥ 0.100 were considered as positive for the detection of antigen.

### 2.4. Determination of Ig Class and Sub-Cloning Procedure

The Ig class of mAbs was determined by using antigen-mediated ELISA [46]. The wells of three replicate microtiter plates coated with 20 μg EPS/mL PBS were incubated successively with hybridoma TCS for 1 h; goat anti-mouse IgG1-, IgG2a-, IgG2b-, IgG3-, IgM-, or IgA-specific antiserum (ISO-2, Sigma) diluted 1:3000 (vol:vol) in PBST for 30 min; and rabbit anti-goat peroxidase conjugate (A5420, Sigma) diluted 1:1000 (vol:vol) for a further 30 min. Bound antibody was visualised with TMB substrate as described. Hybridoma cell lines were sub-cloned three times by limiting dilution, and cell lines were grown in bulk in a non-selective medium, preserved by slow freezing in FBS/dimethyl sulphoxide (92:8 vol:vol), and stored in liquid N_2_.

### 2.5. Antibody Purification and Enzyme Conjugation

Hybridoma TCS of mAb JD4 was harvested by centrifugation at 2150× *g* for 40 min at 4 °C, followed by filtration through a 0.8 μM cellulose acetate filter (10462240, GE Healthcare Life Sciences, Little Chalfont, UK). Culture supernatant was loaded onto a HiTrap Protein A column (17-0402-01, GE Healthcare Life Sciences, Little Chalfont, UK) using a P-1 peristaltic pump (18-1110-91, GE Healthcare Life Sciences, Little Chalfont, UK) with a low pulsation flow of 1 mL/min. Columns were equilibrated with 10 mL of PBS, and column-bound antibody was eluted with 5 mL of 0.1 M glycine-HCl buffer (pH 2.5) with a flow rate of 0.5 mL/min. The buffer of the purified antibody was exchanged with PBS using a disposable PD-10 desalting column (17-0851-01, GE Healthcare Life Sciences, Little Chalfont, UK). Following purification, the antibody was sterile-filtered with a 0.24 µm syringe filter (85037-574-44, Sartorius, Göttingen, Germany) and stored at 4 °C. Protein concentrations were determined using a Nanodrop spectrophotometer, with the protein concentrations calculated using a mass extinction coefficient of 13.7 at 280 nm for a 1% (10 mg/mL) IgG solution. Antibody purity was confirmed by SDS-PAGE and gel staining using Coomassie Brilliant Blue R-250 dye (Thermo Fisher Scientific, Loughborough, UK). Protein A-purified mAb JD4 was conjugated to horseradish peroxidase (HRP) for ELISA studies using a Lightning-Link horseradish peroxidase conjugation kit (701-0000, Bio-Techne Ltd., Abingdon, UK) or to alkaline phosphatase (AKP) for Western blotting studies using a Lightning-Link alkaline phosphatase conjugation kit (702-0010, Bio-Techne Ltd., Abingdon, UK).

### 2.6. Determination of Antibody Specificity

For antibody specificity tests, the wells of two replicate microtiter plates were coated with purified EPS antigens at a concentration of 20 μg/mL PBS from different *Mucorales* fungi and from unrelated moulds of clinical importance (Table 1). The wells were assayed by Direct-ELISA using the JD4-HRP conjugate diluted 1:5000 (vol:vol) in PBST for 1 h, followed by TMB substrate solution for 30 min.

### 2.7. Epitope Characterisation by Heat

The heat stability of the JD4 epitope was determined by heating three replicate EPS samples from *Apophysomyces variabilis* strain CBS658.93 at a concentration of 20 μg/mL PBS in a boiling water bath. At 10 min intervals, 50 μL volumes were removed and, after cooling, were transferred to the wells of microtiter plates for assay by Direct-ELISA using the JD4-HRP conjugate as described.

### 2.8. Polyacrylamide Gel Electrophoresis and Western Blotting

Sodium dodecyl sulphate–polyacrylamide gel electrophoresis (SDS-PAGE) was carried out using 4–20% gradient polyacrylamide gels (161-1159, Bio-Rad, Watford, UK) under denaturing conditions. Antigens were separated electrophoretically at 165 V, and pre-stained markers (161-0318, Bio-Rad, Watford, UK) were used for molecular weight determinations. For Western blotting, separated antigens were transferred electrophoretically onto a PVDF membrane (162-0175, Bio-Rad, Watford, UK) for 2 h at 75 V, and the membrane was blocked for 16 h at 4 °C in PBS containing 1% (wt:vol) BSA. Blocked membranes were incubated with JD4-AKP diluted 1:15,000 (vol:vol) in PBS containing 0.5% (wt:vol) BSA (PBSA) for 2 h at 23 °C. Membranes were washed three times with PBS and once with PBST, and bound antibody was visualised by incubation in substrate solution. Reactions were stopped by immersing membranes in dH_2_O, and membranes were then air-dried between sheets of Whatman filter paper.

### 2.9. Antigen Production In Vitro

For antigen production studies, *A. variabilis* CBS658.93 was grown in YNB + G medium [42,43]. Conical flasks (250 mL) containing 100 mL of autoclaved medium were inoculated with spores to a final concentration of 10^3^ spores/mL, and the cultures were incubated at 37 °C with shaking (120 RPM) in a New Brunswick orbital shaker. At 24 h intervals, three replicate flasks were harvested, and culture fluids were separated from mycelium by filtration through Miracloth (475855-1R, Merck Millipore, Poole, UK). The mycelial biomass was dried for 4 d at 80 °C and weighed. Protein precipitates were prepared by mixing culture filtrates with ethanol at a ratio of 1:4 (vol:vol) in 50 mL Falcon tubes (62.547.254, Sarstedt AG, Nümbrecht, Germany) and chilling them at −20 °C for 16 h. After centrifugation for 10 min at 3200× *g*, the clear supernatants were aspirated, the pellets re-suspended in 1 mL PBS, and the suspensions then stored at −20 °C. On thawing, any insoluble material was removed by centrifugation for 5 min at 16,000× g and then assayed by Direct-ELISA with JD4-HRP conjugate as described.

### 2.10. Culture-Based Detection and Differentiation of Mucorales Fungi

Two replicate 6 cm diameter culture plates (628161, Greiner, Stonehouse, UK) containing MEA were inoculated centrally with 5 mm diameter discs of mycelium generated, using a #2 cork borer, from MEA cultures of *Mucorales* spp. and the non-sporulating mutant (strain ΔAf*brl*A7 [47]) of *Aspergillus fumigatus* (Table 2). The culture plates were incubated at 37 °C for 24 h, after which five discs of mycelium, generated using the cork borer (Figure 1, Step 1, and Appendix A), were transferred to two replicate 25 mL sterile Universal tubes each containing 10 mL of YNB + G medium. The tubes were incubated at 37 °C for 24 h with horizontal shaking (60 RPM), after which the culture fluids were separated from the fungal microcolonies (Appendix A) by filtering through Miracloth. Protein precipitates were prepared by mixing the culture fluids with ethanol at a ratio of 1:4 (vol:vol) in 50 mL Falcon tubes and chilling them at −20 °C for 7 h. The mixtures were centrifuged at 3200× *g* for 10 min, the pelleted material resuspended in 1 mL PBS, and 50 μL volumes of suspension used to coat the wells of microtiter plates for assay by Direct-ELISA using the JD4-HRP conjugate as described. For the detection of *Aspergillus* antigen in Direct-ELISA, the *Aspergillus*-specific mAb JF5 [48] was directly conjugated to HRP as described and used at a concentration of 1:5000 (vol:vol) in PBST. The hyphal biomass retained on Miracloth filters was heated in an oven at 4 d at 80 °C and its dry weight determined.

For lateral-flow immunoassay, solubilised antigens were prepared from the two replicate 24 h old MEA cultures following removal of the mycelial discs (Figure 1, Step 1). Sterile disposable sampling (nasal) swabs (Type G-015, Jiangsu HanHeng Medical Technology Co. Ltd., Changzhou, China) were wetted in 250 μL of TG11-LFD running buffer (RB; PBS containing 0.05% (vol:vol) Tween-20 and 0.05% (vol:vol) Octoxinol-10) contained in two replicate 1.5 mL microcentrifuge tubes. The wetted swabs were used to rub the surfaces of the 24 h old MEA cultures (Figure 1, Step 1) and then returned to the tubes containing the RB for recovery of soluble antigen (Figure 1, Step 2). Antigen extracts were then centrifuged for 5 min at 17,000× *g* (Figure 1, Step 3), and 100 μL of the supernatant was applied to two replicate TG11-LFD tests (Figure 1, Step 4). After 30 min, the intensities of the test (T) and control (C) lines were determined in artificial units (a.u.) using a Cube reader [42,43].

### 2.11. Statistical Analysis

Numerical data were analysed using the statistical programme Minitab (Minitab 16; Minitab, Coventry, UK). Analysis of variance (ANOVA) was used to compare means, and post hoc Tukey–Kramer analysis was then performed to determine statistical significance.

## 3. Results

### 3.1. Production of Hybridomas and mAb Isotyping

A single hybridoma fusion was performed, and 327 hybridoma cell lines were tested in Indirect-ELISA tests for recognition of the immunogen. Seven cell lines produced EPS-reactive antibodies, five of which produced mAbs of the immunoglobulin class G1 (IgG1). The remaining two cell lines produced mAbs of the class G2b (IgG2b). The cell line JD4 (an IgG1) was selected for further evaluation due to its specificity for *Apophysomyces* species and lack of cross-reactivity with non-*Mucorales* fungi.

### 3.2. Specificity of mAb JD4 and Epitope Characterisation

In Direct-ELISA tests of purified EPSs from a range of clinically relevant *Mucorales* and non-*Mucorales* mould pathogens, mAb JD4 was found to be specific to fungi in the *Mucorales* genus *Apophysomyces* (Table 1). No cross-reactivity was found with the other *Mucorales* pathogens tested (*Cunninghamella*, *Lichtheimia*, *Mucor*, *Rhizomucor*, *Rhizopus*, and *Saksenaea*) or with unrelated moulds of clinical importance (*Aspergillus*, *Fusarium*, *Lomentospora*, and *Scedosporium* spp.) [49]. Direct-ELISA tests showed that mAb JD4 binds to a heat-labile epitope (Figure 2A), present in a single 15 kDa *Apophysomyces*-specific protein in Western immunoblots but absent in purified EPS preparations from other *Mucorales* fungi and from *Aspergillus fumigatus* (Figure 2B,C).

### 3.3. Antigen Production In Vitro

Hyphal growth of *A. variabilis* CBS658.93 in YNB + G shake culture plateaued at 24 h post-inoculation with a mean dry weight of 148.93 mg ± 0.020 mg (Figure 3A). Antigen production, determined by Direct-ELISA, peaked after 24 h and declined thereafter until the end of the experimental period at 120 h (Figure 3B). In Western immunoblots, a 15 kDa immunoreactive antigen was first discernible at 24 h and was detectable throughout the experimental period (Figure 3 C). A second immunoreactive band of ~35 kDa was also evident, possibly as a dimer of the 15 kDa protein.

### 3.4. Culture-Based Detection and Differentiation of Mucorales Fungi

#### 3.4.1. Lateral-Flow Immunoassay

Using crude antigen extracts from swabs of 24 h old MEA cultures of the fungi, the TG11-LFD test was shown to be rapid and specific, detecting only the non-sporulating *Mucorales* species of *Apophysomyces* and *Saksenaea* and a strain of *Lichtheimia ramosa* (DRH226533346) recovered from a biopsy of cutaneous mucormycosis. Antigen extracts from these *Mucorales* fungi gave mean test (T) line Cube reader values of <40 a.u. (Table 2), below the threshold a.u. for test positivity (≤400 a.u.). Extracts from the non-sporulating mutant of *Aspergillus fumigatus* (strain ΔAf*brl*A7) gave a mean T line a.u. value of 474.0 ± 19.0, showing a lack of detection of this pathogen. The negative controls (RB only and purified EPS from *A. fumigatus* Af293 at a concentration of 100 μg/mL RB), were similarly negative with mean T line a.u. values > 400 a.u. The positive control comprising purified EPS from the *Lichtheimia corymbifera* strain CBS109940 at a concentration of 100 μg/mL RB gave a positive test result, with a mean T line value of 29.5 ± 2.2.

#### 3.4.2. Direct-ELISA

In Direct-ELISA tests of protein precipitates from fungi grown for 24 h in YNB + G shake cultures (Figure 4A and Appendix A), mAb JD4 was shown to be genus-specific, reacting with *Apophysomyces* species only (Figure 4B). The *Aspergillus*-specific mAb JF5, acting as an immunoglobulin G (IgG) control, reacted with *A. fumigatus* Af293 only (Figure 4C).

## 4. Discussion

Mucormycosis is an aggressive angio-invasive disease of humans, manifesting as life-threatening rhino-orbital-cerebral, pulmonary, gastro-intestinal, cutaneous, and disseminated infections [41]. Despite best-practice treatment, the overall all-cause mortality remains high at approximately 50% [50]. While cutaneous disease is associated with lower mortality of around 30%, extensive surgery including limb amputation is frequently required to control the infection. Due to the rapid progression and destructive nature of mucormycosis, early diagnosis is crucial to improving mortality and limiting tissue loss and disfigurement [4,13].

Options for the treatment of mucormycosis with antifungal drugs are limited [51,52]. It is imperative that *Mucorales* pathogens are differentiated from *Aspergillus fumigatus* due to the intrinsic resistance of *Mucorales* species to short-tailed azoles, such as voriconazole. The latter is used as a front-line antifungal drug for aspergillosis [53], the most frequent differential diagnosis associated with mucormycosis [54]. Furthermore, discrimination of *Apophysomyces* from *Sakasenaea* species is required, since *Apophysomyces* spp. display decreased sensitivity to amphotericin B [55,56], the antifungal drug most effective for treatment of mucormycosis [4,57].

Currently, diagnosis of cutaneous mucormycosis requires histochemistry and culture from a tissue biopsy [4,58], with identification of characteristic morphological features (broad ribbon-like aseptate hyphae and spore-bearing structures) remaining the gold-standard procedure for the detection and differentiation of infecting species. However, *Apophysomyces* and *Saksenaea* species are notorious in their failure to sporulate on standard mycological media, preventing accurate identification and delaying appropriate treatment. To mitigate this, we report here the development of simple culture-based detection methods that incorporate *Mucorales*-specific immunoassays for the detection and discrimination of these fastidious pathogens. Rapid discrimination of *Mucorales* species from *A. fumigatus* is achieved with lateral-flow immunoassay, with subsequent differentiation of non-sporulating species of *Apophysomyces* and *Saksenaea* using a newly developed mAb, JD4, in ELISA.

The pan-*Mucorales*-specific monoclonal antibody (mAb), TG11, used in the lateral-flow immunoassay has been used previously to detect *Mucorales* invasive hyphae in lung samples using immunohistochemistry [44] and in a lateral-flow device (LFD) to detect the *Mucorales* EPS antigen in bronchoalveolar lavage (BAL) samples from patients with pulmonary mucormycosis [45]. In a competitive LFD format such as the pan-*Mucorales*-specific TG11-LFD [43], the test (T) line signal intensity is inversely proportional to the analyte concentration [59]. Consequently, a strong test line signal means that little or none of the target analyte (*Mucorales* EPS) is present in the sample. TG11-LFD tests using MEA culture swabs of the *Mucorales* pathogens resulted in test line “whiteouts” (complete absence of visible test line signals), indicating positive test results (presence of the EPS analyte in swab samples). This equated to Cube reader values in the range of 15 to 40 artificial units (a.u.). Cube reader values for swabs from the non-sporulating mutant of *A. fumigatus*, ΔAf*brl*A7 [47], and for *A. fumigatus* purified EPS, exceeded the threshold value for test positivity of ≥400 a.u. [43], further demonstrating the specificity of the TG11-LFD for *Mucorales* fungi. The simple and quick LFD swab test was therefore able to accurately differentiate non-sporulating *Apophysomyces* and *Saksenaea* species from the most common mould pathogen of humans, *A. fumigatus*. This is especially important for skin and soft tissue infections where the symptoms of cutaneous aspergillosis can overlap with those of cutaneous mucormycosis [58], or where mixed *Mucorales* and *Aspergillus* infections are present [14,60]. The TG11-LFD swab test was also able to rapidly detect a strain of *Lichtheimia ramosa* recovered from culture of a tissue biopsy from a patient with cutaneous mucormycosis. Consequently, this swab test is applicable to other *Mucorales* fungi capable of causing infections in humans, including those similarly capable of causing necrotising fasciitis such as *Lichtheimia* [61,62]. The LFD test conforms to the ASSURED (affordable, sensitive, specific, user-friendly, rapid, equipment-free, delivered) criteria for diagnostics for the developing world [63], where access to more sophisticated and costly diagnostic tests such as qPCR [64], MALDI-TOF [38], or metagenomic next-generation sequencing (NGS) [65] is limited or absent.

The genera *Apophysomyces* and *Saksenaea* are ascribed to the family Saksenaeaceae within the order *Mucorales*, with both considered species complexes [66]. Despite their close molecular taxonomic relatedness [67], mAb JD4 was able to discriminate between the two complexes, binding to a major 15 kDa antigen specific to the genus *Apophysomyces*. The JD4 antigen was readily detected in protein precipitates from all four *Apophysomyces* species reported as human pathogens [13], comprising *A. elegans* [68], *A. mexicanus* [69], *A. ossiformis* [70,71,72], and *A. variabilis* [8,10,16,19,22,24,25,39,73], but did not cross-react with species of *Saksenaea*, notably the human pathogens *S. dorisiae* [36], *S. erythrospora* [74,75,76,77], and *S. vasiformis* [9,11,23,78]. We were unable to determine the reactivity of mAb JD4 with *A. trapeziformis* [26,79] due to the lack of availability of this species for testing.

Enzyme-linked immunosorbent assay (ELISA) is a well-established and commonly used immunoassay performed in most diagnostic laboratories worldwide. The commercial availability of mAb JD4 means that diagnostic facilities equipped for this technique can use the mAb-based ELISA to detect and differentiate *Apophysomyces* species from other fungal pathogens that cause rapidly progressive cutaneous and soft tissue mycoses in humans. When this is combined with the rapid TG11-LFD test, improvements are offered in the speed, sensitivity, and specificity of *Mucorales* detection based on mycological culture, a gold-standard procedure for mucormycosis detection [4,52].

## Figures and Tables

**Figure 1 antibodies-14-00085-f001:**
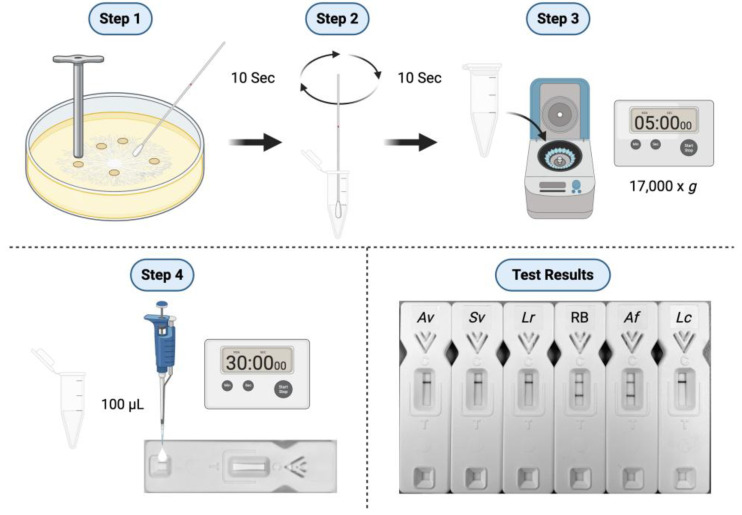
The procedure for the detection of pan-*Mucorales*-specific antigen by lateral-flow immunoassay (LFIA). Step 1. Fungus is grown at 37 °C for 24 h on MEA, and five discs of mycelium, prepared using a cork borer, are transferred to YNB + G shake culture. The remaining MEA culture is then swabbed for 10 sec with a disposable sampling swab wetted with TG11-LFD running buffer (RB). Step 2. Soluble antigen is recovered from the swab by rotation for 10 sec in 250 μL RB contained in a 1.5 mL micro-centrifuge tube. Step 3. The tube is centrifuged for 5 min at 17,000× *g* and the supernatant removed for testing by TG11-LFD. Step 4. One hundred μL of supernatant is added to the sample pad of the LFD, and, after 30 min, the intensities of the test (T) and internal control (C) lines are determined in artificial units (a.u.) using a Cube reader [42,43]. Representative Test Results show complete loss of T lines for swab samples from *Av* (*A. variabilis* CBS658.93), *Sv* (*Saksenaea vasiformis* CBS113.96), and *Lr* (*L. ramosa* DRH226533346), while there is no loss of the T line for *Af* (*Aspergillus fumigatus*, strain ΔAf*brl*A7). The negative control shown here consists of RB only, while the positive control (*Lc*) consists of purified EPS from *Lichtheimia corymbifera* CBS109940 at a concentration of 100 μg/mL RB. The corresponding Cube reader a.u. values for T lines are shown in Table 2.

**Figure 2 antibodies-14-00085-f002:**
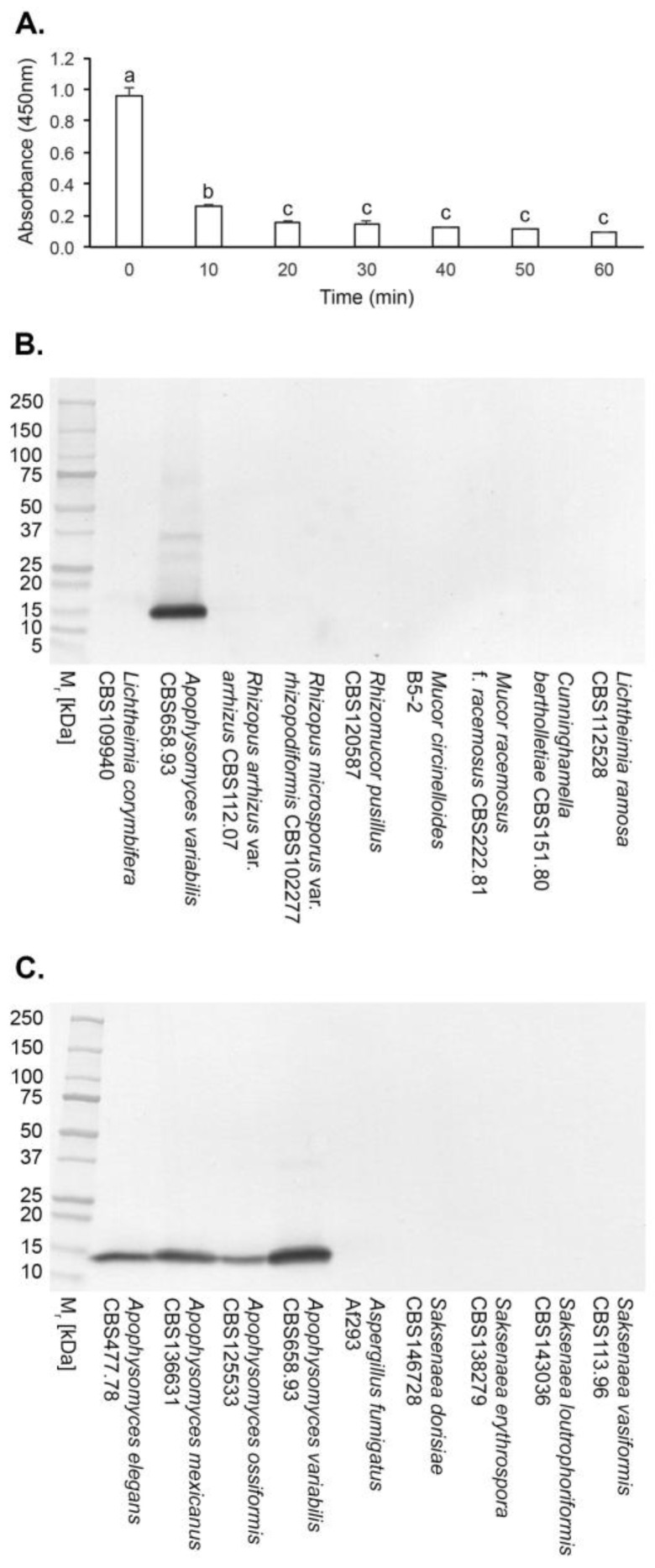
Epitope characterisation by heat treatment and specificity of mAb JD4. (**A**) JD4 Direct-ELISA of the *Apophysomyces variabilis* CBS658.93 EPS antigen following heating at 100 °C for 10, 20, 30, 40, 50, and 60 min. Bars are the means of three replicates ± SEs, and bars with the same letter are not significantly different at *p* < 0.05. After 10 min of heat treatment, there was a significant (*p* < 0.05) decrease in ELISA absorbance value compared to the control (no heat treatment), indicating that the mAb binds to a heat-labile epitope. (**B**) Western immunoblot of EPS antigens from *Mucorales* fungi of clinical importance, showing a major JD4 immunoreactive protein of 15 kDa specific to *Apophysomyces variabilis*. (**C**) Western immunoblot of purified EPSs prepared from species of the *Mucorales* fungi *Apophysomyces* and *Saksenaea* and from the fungus *Aspergillus fumigatus*. Immunoreactive proteins of 15 kDa are present in the purified EPS preparations of *Apophysomyces* spp. but absent in *Saksenaea* spp. and *Aspergillus fumigatus* preparations.

**Figure 3 antibodies-14-00085-f003:**
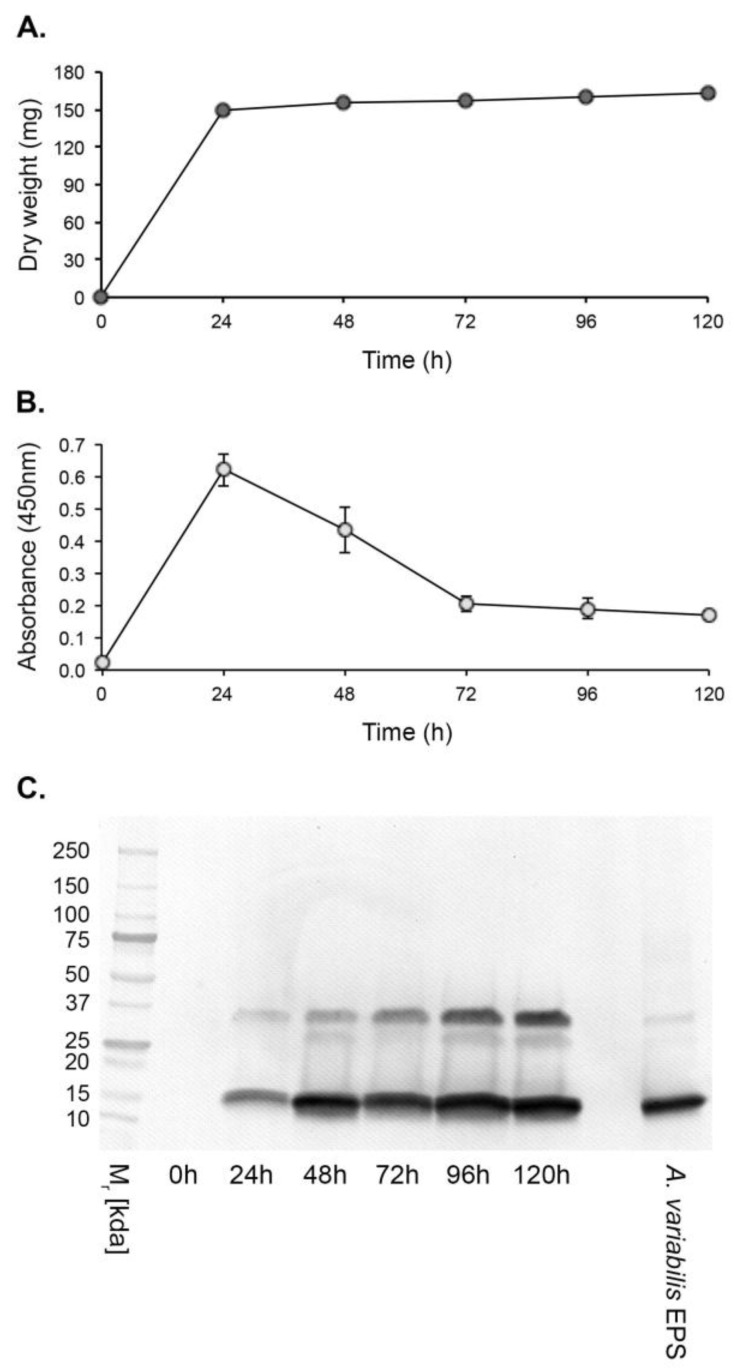
In vitro production of the JD4 antigen by *A. variabilis* CBS658.93. (**A**) Dry weights of the pathogen over the 120 h experimental period. (**B**) Direct-ELISA of YNB + G protein precipitates using mAb JD4. Each data point is the mean of three replicates ± SE (**B**,**C**). (**C**) Western immunoblots of protein precipitates, showing the presence of a 15 kDa immunoreactive protein and a higher-molecular-weight species of ~35 kDa. The positive control consisted of purified *A. variabilis* EPS antigen at a concentration of 80 ng/mL.

**Figure 4 antibodies-14-00085-f004:**
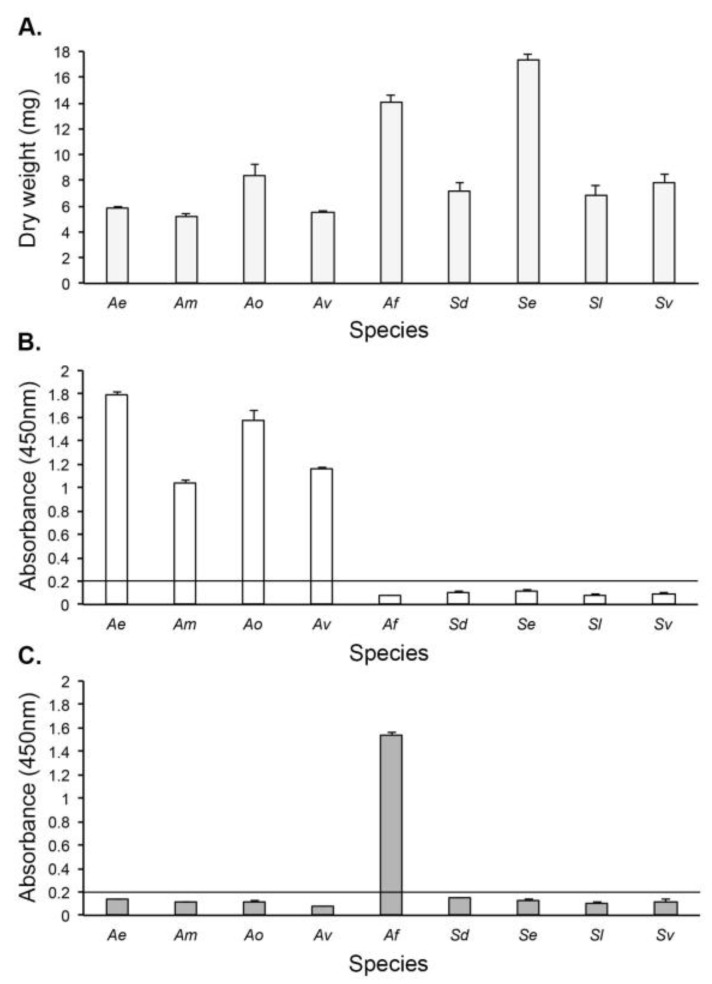
In vitro production of the JD4 antigen by species of *Apophysomyces*, *Saksenaea*, and *Aspergillus*: *Ae* (*Apophysomyces elegans*), *Am* (*Apophysomyces mexicanus*), *Ao* (*Apophysomyces ossiformis*), *Av* (*Apophysomyces variabilis*), *Af* (*Aspergillus fumigatus*, strain ΔAf*brl*A7), *Sd* (*Saksenaea dorisiae*), *Se* (*Saksenaea erythrospora*), *Sl* (*Saksenaea loutrophoriformis*), and *Sv* (*Saksenaea vasiformis*). (**A**) Dry weights of hyphal microcolonies after 24 h of growth in YNB + G shake cultures. (**B**) Direct-ELISA of protein precipitates using the *Apophysomyces*-specific mAb JD4. The threshold ELISA absorbance value for immunoassay positivity is ≥0.2, with values above this threshold positive for the detection of *Apophysomyces* antigen. (**C**) Direct-ELISA of protein precipitates using the *Aspergillus*-specific mAb JF5 [49]. The threshold ELISA absorbance value for immunoassay positivity is ≥0.2, with values above this threshold positive for the detection of *Aspergillus* antigen. Each bar is the mean of three replicates ± SE (**A**–**C**).

**Table 1 antibodies-14-00085-t001:** Details of human pathogenic fungi used to determine the specificity of mAb JD4, and results of Direct-ELISA tests against purified EPS antigens.

Species	Isolate Number	Source ^1^	JD4 Direct-ELISA Absorbance (450 nm) ^2^
*Apophysomyces elegans*	477.78	CBS	1.466 ± 0.005
*Apophysomyces mexicanus*	136361	CBS	1.224 ± 0.006
*Apophysomyces ossiformis*	125533	CBS	0.674 ± 0.018
*Apophysomyces variabilis*	658.93	CBS	1.646 ± 0.022
*Aspergillus fumigatus*	Af293	FGSC	0.082 ± 0.003
*Cunninghamella bertholletiae*	151.80	CBS	0.084 ± 0.011
*Fusarium solani*	224.34	CBS	0.049 ± 0.000
*Lichtheimia corymbifera*	109940	CBS	0.093 ± 0.002
*Lichtheimia ornata*	142195	CBS	0.083 ± 0.000
*Lichtheimia hyalospora*	146576	CBS	0.073 ± 0.002
*Lichtheimia ramosa*	112528	CBS	0.059 ± 0.003
*Lichtheimia ramosa*	124197	CBS	0.071 ± 0.005
*Lichtheimia ramosa*	2845	NCPF	0.067 ± 0.002
*Lomentospora prolificans*	3.1	CRT	0.056 ± 0.001
*Mucor circinelloides*	B-52	CRT	0.050 ± 0.001
*Mucor racemosus* f. *racemosus*	222.81	CBS	0.076 ± 0.003
*Rhizomucor pusillus*	120587	CBS	0.073 ± 0.002
*Rhizopus arrhizus*	111233	CBS	0.050 ± 0.001
*Rhizopus arrhizus* var. *arrhizus*	112.07	CBS	0.044 ± 0.002
*Rhizopus microsporus* var. *rhizopodiformis*	102277	CBS	0.039 ± 0.005
*Saksenaea dorisiae*	146728	CBS	0.044 ± 0.001
*Saksenaea erythrospora*	138279	CBS	0.044 ± 0.002
*Saksenaea loutrophoriformis*	143036	CBS	0.045 ± 0.006
*Saksenaea vasiformis*	113.96	CBS	0.056 ± 0.001
*Scedosporium apiospermum*	8353	CBS	0.045 ± 0.001
*Scedosporium aurantiacum*	121926	CBS	0.062 ± 0.003
*Scedosporium boydii*	835.96	CBS	0.057 ± 0.003

^1^ CBS, Westerdijk Fungal Biodiversity Institute, The Netherlands; CRT, C.R. Thornton; FGSC; Fungal Genetics Stock Center, Kansas City University, USA. ^2^ For Direct-ELISA tests with mAb JD4, the absorbance values are the means of two replicates ± SEs, and the threshold absorbance value for immunoassay positivity is ≥0.100.

**Table 2 antibodies-14-00085-t002:** Details of human pathogenic fungi and results of TG11-LFD tests using swabs from fungal cultures grown for 24 h on MEA medium.

Species	Isolate Number	Source ^1^	TG11-LFD (a.u.) ^2^
*Apophysomyces elegans*	477.78	CBS	34.5 ± 1.5
*Apophysomyces mexicanus*	136361	CBS	27.8 ± 0.1
*Apophysomyces ossiformis*	125533	CBS	30.8 ± 0.3
*Apophysomyces variabilis*	658.93	CBS	15.5 ± 4.5
*Aspergillus fumigatus* ΔAf*brl*A7	A1176	FGSC	474.0 ± 19.0
*Lichtheimia ramosa* ^3^	226533346	DRH	15.7 ± 1.5
*Saksenaea dorisiae*	146728	CBS	38.9 ± 0.5
*Saksenaea erythrospora*	138279	CBS	34.4 ± 1.7
*Saksenaea loutrophoriformis*	143036	CBS	21.0 ± 2.8
*Saksenaea vasiformis*	113.96	CBS	22.5 ± 1.0
Negative Control ^4^	Af293	FGSC	591.7 ± 79.9
Positive Control ^5^	109940	CBS	29.5 ± 2.2
RB only	-	-	506.9 ± 20.8

^1^ CBS, Westerdijk Fungal Biodiversity Institute, The Netherlands; CRT, C.R. Thornton; FGSC; Fungal Genetics Stock Center, Kansas City University, USA. ^2^ For TG11-LFD tests, the T line intensity measured in artificial units (a.u.) is the mean of two replicate samples ± SE, and the threshold a.u. value for test positivity is ≤400 [43]. ^3^ Isolate cultured from the hand of a patient with cutaneous mucormycosis. ^4^ Negative control comprising purified EPS from *A. fumigatus* strain Af293 at 100 μg/mL running buffer (RB). ^5^ Positive control comprising purified EPS from *L. corymbifera* strain CBS109940 at 100 μg/mL RB. All tests had internal control (C) line intensities of ≥400 a.u.

## Data Availability

The data presented in this study are available on request from the corresponding author but are not publicly available due to commercial confidentialities. The monoclonal antibody JD4 and the TG11-LFD test are available from ISCA Diagnostics Limited.

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
