# Peer review of "Monoclonal Antibodies Can Aid in the Culture-Based Detection and Differentiation of Mucorales Fungi—The Flesh-Eating Pathogens Apophysomyces and Saksenaea as an Exemplar"

_2073-4468, 2025, doi:10.3390/antib14040085_

Round 1

Reviewer 1 Report

Comments and Suggestions for Authors

The authors present the development and characterization of the monoclonal antibody JD4, which specifically recognizes Apophysomyces species within the Mucorales, thereby addressing an important diagnostic challenge posed by non-sporulating isolates. A key innovation is the demonstration that the JD4-ELISA, when combined with the pan-Mucorales TG11 lateral flow assay, enables rapid, sensitive, and specific differentiation of clinically relevant pathogens such as Apophysomyces and Saksenaea, with clear implications for guiding timely and appropriate antifungal therapy. The integration of simple culture-based detection methods with antibody-based immunoassays represents a significant advance, particularly for use in resource-limited settings.

Revisions requested:

Introduction

  • Line 33: It may be clearer to refer directly to theMucorales order, as current mycological literature discourages use of “Zygomycetes” as a class.

Materials and Methods

  • Line 84: Please provide a brief description of the method for inducing Saksenaea sporulation, rather than citing only.
  • Line 87: Similarly, please add a short description of EPS preparation.
  • Consider separating the description of hybridoma production and ELISA screening into distinct subsections to improve clarity.
  • For the ELISA experiments and elsewhere as appropriate, please specify the number of replicates performed.
  • Line 121: The centrifugation speed is reported very precisely as 2,147 × g. Is there a specific reason for this value, rather than rounding?
  • Western blotting: The blocking step of 16 h is unusually long (commonly 1–2 h). Please clarify whether this extended incubation is required for technical reasons.

Results

  • Section 3.2: Instead of “Figure 1A, and 1B,C,” it appears the authors intended “Figure 2A, and 2B,C.”

Discussion

  • Line 346: Please add supporting references for the claims regarding the current diagnosis of cutaneous mucormycosis.
  • Line 370: Correct spelling of Apophysomyces.
  • Line 390: Correct spelling of Saksenaea.

Reviewer 2 Report

Comments and Suggestions for Authors

G. E. Davies and C. R. Thornton submitted a manuscript entitled ‘Monoclonal antibodies can aid in the culture-based detection and differentiation of Mucorales fungi - the flesh-eating pathogens Apophysomyces and Saksenaea as an exemplar’ to be considered for its publication in the journal Antibodies. Thus, after having read carefully the manuscript, I consider it suitable for its publication after a minor revision. First of all, the topic is very interesting and important. This reviewer agrees with the authors ‘the number of infections caused by Mucorales fungi is increasing dramatically worldwide. Then, all efforts behind their understanding are welcome, of course including novel methods of early detection. I think this is the main strength of the manuscript by Davies. Besides, the manuscript is well written and easy to read. Introduction shows suitably the context of the work. References cited are many but pertinent. Materials and methods section is described in detail, making the work reproducible by others. Figures and plots are fine (albeit adding colors would be acknowledged). Results are masterfully discussed (worthy of note). However, a minor revision would enhance the manuscript.

It would have been good to add the chemical structures of common antifungal drugs to treat Mucorales fungi infections, especially those against  Apophysomyces and Saksenaea. (a new Fig. after line 47 (introductory part of the manuscript).

Related to the previous point, I respectfully suggest authors add catching-eye images to complete the introductory part. For instance, some pathogenic fungi microscope photographs and/or necrotized tissues (beware of relevant ethical principles and sensitivity to disturbing images!!).

Suppl. Fig. 1 and Figs 2B, 2C and 3C should be placed in the main article.

A conclusions section or at least a conclusions paragraph is missing, for instance to contrast the hypothesis, and to highlight the main findings in a summarized and subjective manner.
